# Effects of Heat Treatment on the Microstructure and Properties of a Cast Nickel-Based High-Cr Superalloy



**Hongguo Lu** [1,2], **Minghui Yang** [1], **Li Zhou** [1,*], **Zhonggang Ma** [2], **Bin Cui** [2], **Fengshi Yin** [3,*] and **Daoqian Li** [2]

1    School of Chemistry and Chemical Engineering, Shandong University of Technology, Zibo 255049, China
2    Shandong Ruitai New Material Technology Co., Ltd., Zibo 256100, China
3    School of Mechanical Engineering, Shandong University of Technology, Zibo 255049, China
*    Correspondence: zhouli@sdut.edu.cn (L.Z.); fsyin@sdut.edu.cn (F.Y.);
     Tel.: +86-1516-922-6899 (L.Z.); +86-1360-533-1243 (F.Y.)

**Abstract:** The effect of solution treatment and intermediate heat treatment on the microstructure and properties of a new cast nickel-based high-Cr superalloy was investigated in this paper. The results indicate that the tensile strength and elongation at 900 °C increase when the solution temperature increases from 1160 °C to 1180 °C and then decrease when the solution temperature changes from 1180 °C to 1200 °C and 1220 °C. The stress rupture test results of the high-Cr superalloy under conditions of 900 °C/275 MPa shows that the rupture time, elongation, and reduction of area initially increased and then decreased with the increase in solution treatment temperatures. The results of stress rupture tests for the alloy after intermediate heat treatment followed by furnace-cooling, air-cooling, and water-cooling show that the morphology and distribution of $\gamma'$ phase have a great influence on the tensile test results at 900 °C of the alloy but no obvious influence on the test at 900 °C/275 MPa. The microstructure analysis of the superalloy after heat treatment shows that: when the solution treatment temperatures are at 1200 °C and 1220 °C, the incipient melting appears in the interdendritic region, which can severely deteriorate mechanical properties; the morphology of $\gamma'$ phase changes gradually from cube to spherical; and a large number of fine $\gamma'$ phase precipitates in the $\gamma$ channel are found with increasing cooling rate after intermediate heat treatment.

**Keywords:** nickel-based high-Cr superalloy; solution treatment; eutectic structure; intermediate treatment

## 1. Introduction

The nickel-based superalloy was widely used in gas turbines due to its excellent resistance to corrosion and oxidation under severe high-temperature conditions [1]. The different element ratios and heat-treatment conditions have significant influence on the properties of the alloy. Therefore, there is still a lot of room to improve the performance of superalloy turbines. To further upgrade the strength and resistance to corrosion and oxidation of superalloy, the strengthening process or mechanism has been extensively investigated.

According to previous studies, the different elements ratios and heat treatment temperature can influence the lattice constants of the matrix and precipitated phase and then affect the properties of the alloy [2–7]. The thermal corrosion resistance and high-temperature oxidation resistance of superalloy can be improved by adding trace grain boundary strengthening elements of B and Zr or refractory elements such as W, Mo, Nb, and Hf [2]. Changes in the microstructure by a suitable heat treatment process can further improve the performance of the alloy. As is well known, the microstructure of casting nickel-based superalloy consists of the $\gamma'$ phase, carbides, borides, nitrides, TCP phases, $\gamma/\gamma'$ eutectic structure, and other precipitated phases. The heat-treatment process often includes three steps: the solid solution treatment, the intermediate treatment, and the aging treatment. The research shows that the cast structure of a coarse $\gamma'$ phase, the carbides, and $\gamma/\gamma'$ eutectic can be dissolved in the matrix through the high-temperature

solution treatment [3–6], while the following intermediate and aging treatments contribute to the formation of a fine $\gamma'$ phase [7]. Therefore, the heat treatment plays an important role in controlling the shape, size, distribution, and volumes of $\gamma'$ phase, $\gamma/\gamma'$ eutectic, and carbides.

In addition, the solid-solution treatment can also improve the elements' segregation. It promotes the separation of W elements in dendrites and diffusion to interdendrites. Moreover, elements of Hf, Nb, Ti, and Cr diffused into the dendrites, the segregation degree of each component of the alloy was greatly reduced, and the mechanical properties of the alloy were improved [8]. However, the excessively high solution treatment temperature will cause the "incipient melting" phenomenon [9,10]. Incidentally, the cooling has an important influence on the $\gamma'$ phase morphologies, and with decreases in the cooling rate, the $\gamma'$ phase cubic degree increases [11,12].

Recently, most work was focused on the relationship between deformation behavior and the precipitates of the cast nickel-based superalloys after heat treatment. L. Shi et al. [13–15] found that the deformation was dominated by dislocation shearing in the second phase when the temperature was low, and in contrast, the dislocations by passing the second phase when on the high temperature. Jian Wang [16] researched the primary MC decomposition and its effects on the tensile test behaviors at 900 °C in a high-Cr nickel-based superalloy when the solution treatment temperature was 1160 °C and found that the cracks mainly initiated in the grain boundaries, while during the stress rupture test at 900 °C/274 MPa, the cracks merely formed in the primary MC decomposition region. Yuan et al. [17] found that the alloy after 1050–1150 °C solution treatment did not have significant changes in the morphology of grain and grain boundary, but with an increase in the solution treatment temperature, the $\gamma'$ phase of the interdendritic region becomes cubic gradually; simultaneously, the secondary $\gamma'$ phase is precipitated in the $\gamma$ channel. Mao et al. [18] found the alloy composition was of 8% Cr, 25% Mo and 62% Ni and found that the TCP phase precipitated in Re 5% alloy was σ and in Re 10% were σ and P phases, and that the size of TCP-phases increases and the volume fraction of the TCP phases decreases with the increase in the heat-treatment temperatures.

In summary, the low-heat treatment temperature (≤1160 °C) has a great influence on the precipitation phase and mechanical properties of the alloy. However, the alloy in use often suffers the high-temperature heat treatment phenomenon and severely deteriorated mechanical properties, and the effect of high heat-treatment temperature on high-Cr superalloy precipitates has not been studied in detail. Therefore, it is necessary to systematically investigate the mechanism whereby high-temperature heat treatment influences the microstructure and properties of high-Cr nickel-based superalloy.

In this paper, a type of high-Cr superalloy is designed which contains many solid solution-strengthening elements aimed at the gas turbines. Moreover, the influences of the high-heat treatment process on the microstructure, the behavior of deformation, and the mechanism of the second $\gamma'$ phase of the high-Cr superalloy have been investigated and discussed in detail. Finally, this work has provided an experimental basis for an appropriate heat-treatment schedule.

## 2. Materials and Methods

The nominal composition of the high-Cr superalloy used in this study is listed in Table 1. The high-Cr superalloy was an alloy improved by K444. The master alloy ingots were cut and re-melted at 1580 °C for 5 min in a ZGJL0.5-100-2.5 vacuum induction furnace, then poured into investment molds with back-filling sand preheated to 900 °C and cast bar-shaped specimens for further research. In order to research the effect of heat-treatment on the properties of the high-Cr superalloy, the specimens were heat-treated at different solid-solution temperatures and cooling conditions after intermediate heat treatment. The heat-treatment regimens were summarized in Table 2. The intermediate heat treatments were followed by furnace-cooling (FC), air-cooling (AC), and water-cooling (WC). All heat

treatment experiments were respectively carried out in a muffle furnace and under the air atmosphere.

**Table 1.** Chemical composition of the high-Cr superalloy ingots (*wt*,%).

| C | Cr | Co | W | Mo | Al | Ti | Nb | Hf | B | Zr | Ni |
|------|------|------|------|-----|-----|-----|-----|-----|------|-----|-------|
| 0.06 | 15.8 | 10.7 | 5.16 | 2.0 | 3.1 | 4.5 | 0.2 | 0.3 | 0.06 | 0.2 | 57.92 |

**Table 2.** Summarized heat-treatment conditions of high-Cr Superalloy.

| Sample Number | Solid Solution Treatment | Intermediate Treatment | Aging Treatment |
|---|---|---|---|
| $S_1$ | 1160 °C + 4 h + AC | 1050 °C + 14 h + AC | |
| $S_2$ | 1180 °C + 4 h + AC | 1050 °C + 14 h + AC | |
| $S_3$ | 1200 °C + 4 h + AC | 1050 °C + 14 h + AC | 750 °C + 12 h + AC |
| $S_4$ | 1220 °C + 4 h + AC | 1050 °C + 14 h + AC | |
| $S_5$ | 1180 °C + 4 h + AC | 1050 °C + 14 h + FC | |
| $S_6$ | 1180 °C + 4 h + AC | 1050 °C + 14 h + WC | |

The standard specimens for the tensile test with a diameter of 5 mm and a gauge length of 25 mm were cut by Wire Electrical Discharge Machining (WEDM) and machined longitudinally from the heat-treated samples. The tensile tests were carried out at 900 °C on an RMT-J5 mechanical test machine, and the stress rupture tests were carried out on a CSS3905 creep test machine under the conditions of 900 °C/275 MPa. The longitudinal sections were cut by WEDM from the fractured specimens. The specimens for microstructure examination were polished and chemically etched in a solution of 20 g $CuSO_4$ + 100 mL HCl + 5 mL $H_2SO_4$ + 80 mL $H_2O$. SEM observation was performed using a Quanta 250 environmental-scanning electron microscope equipped with energy-dispersive spectroscopy.

## 3. Results

### 3.1. Microstructure after Heat Treatment

Figure 1 shows the microstructure of the high-Cr superalloy heat-treated by S1, S2, and S3 methods. As shown in Figure 1, the microstructure of the alloy was mainly composed of γ matrix, ordered γ′ phase, and carbides. When the solution temperature was 1160 °C, there was still obvious dendrite morphology because the coarse γ′ phase in the interdentritic region was not dissolved completely (Figure 1a). With the increase in the solution temperature up to 1180 °C, the dendrite morphology was not obviously (Figure 1b) indicating that the cast coarse γ′ phase was fully dissolved into the matrix and changed into a fine cubic γ′ phase with an average edge length of 0.25–0.4 µm (Figure 1d). Further increasing the solution temperature up to 1200 °C and 1220 °C, the interdendritic region appears incipient-melting (Figure 1c). High-magnification images show that there are much finer γ′ particles with an average size of 5 nm distributed in the γ channels, as shown in Figure 1d.

It was found that there were two kinds of carbides with different compositions in the heat-treated superalloy, as shown in the BSE image. Figure 2 shows the morphology and EDS spectrum of the carbides in the alloy heat-treated by the S2 method. The EDS results showed that the carbides marked with arrow A in Figure 2a had a script-like morphology and were rich in the Ti element, indicating that it belonged to an MC-type carbide, while the bright phase marked with arrow B was rich in the W element, indicating that it belonged to an $M_6C$-type carbide [2]. It was believed that the $M_6C$-type carbide in superalloy was formed during solution treatment by the following reaction:

$$MC + \gamma \rightarrow M_6C + \gamma'$$

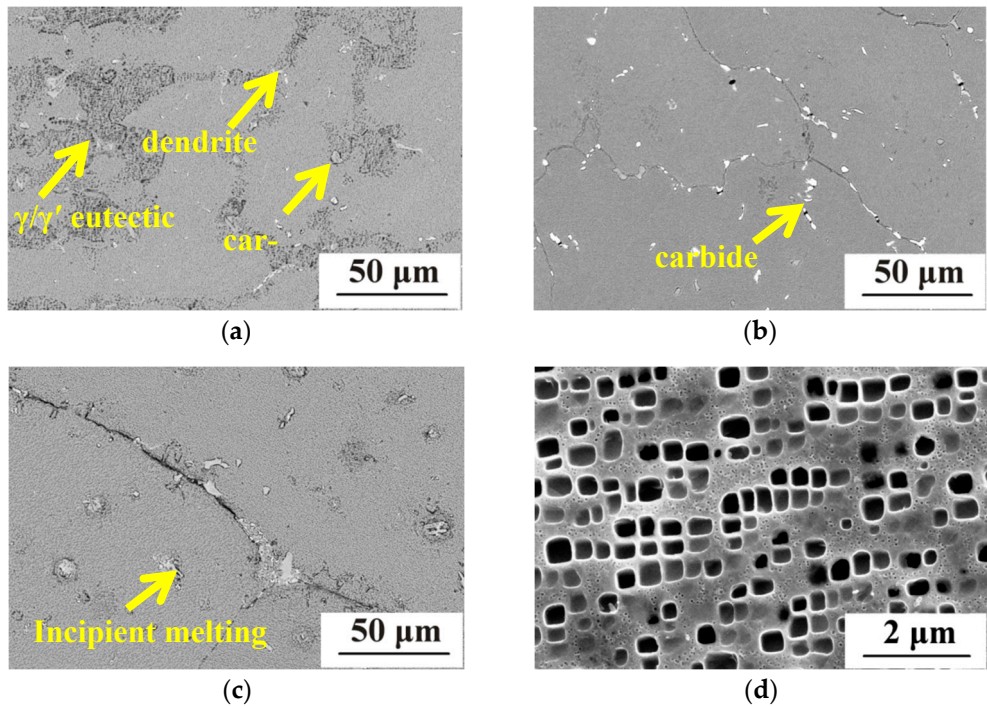

**Figure 1.** The microstructure of high-Cr superalloy with different solution treatment temperatures. (**a**) 1160 °C, (**b**) 1180 °C, (**c**) 1200 °C, (**d**) high magnification images of the high-Cr superalloy in Figure 1b.

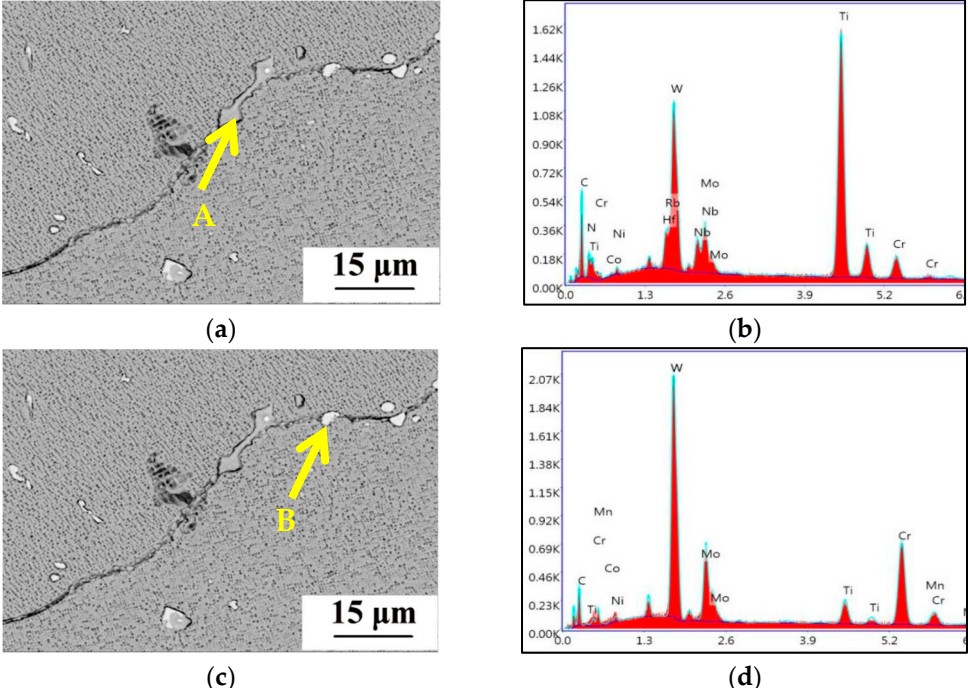

**Figure 2.** The results of carbide morphology and energy-spectrum analysis of high-Cr superalloy after S2 mode (**a,c**) high magnification of the high-Cr superalloy; (**b**) the analysis result of the precipitated phase energy spectrum at point A in figure a; (**d**) the analysis result of the precipitated phase energy spectrum at point B in (**a**).

The microstructure of high-Cr superalloy by S2, S5, and S6 heat-treatment methods was shown in Figure 1d and Figure 3. It was shown that the intermediate treatment cooling rate have an important effect on γ′ phase. When the air cooling and water cooling were

adopted after the intermediate treatment at 1050 °C, the microstructure had two types of $\gamma'$ phase: the larger cubic $\gamma'$ phase and the smaller $\gamma'$ phase in $\gamma$ channel. When the furnace cooling was chosen, only the larger cubic $\gamma'$ phase was on the matrix, indicating that the different cooling rates affected the nucleation and growth of the $\gamma'$ precipitates.

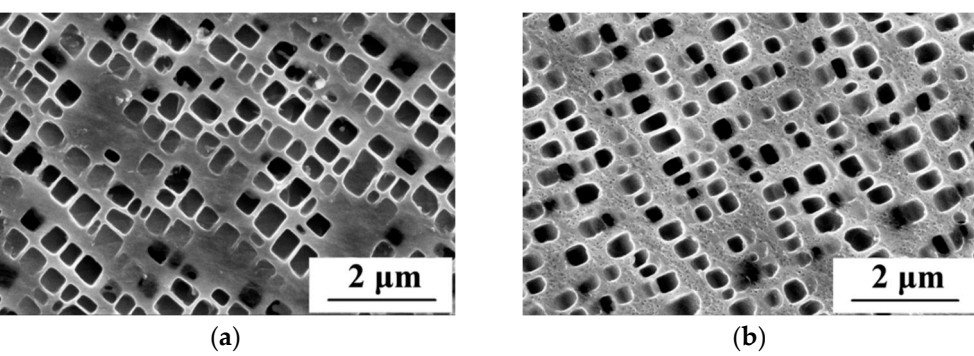

(**a**)   (**b**)

**Figure 3.** The microstructure of high-Cr superalloy with different cooling conditions after the intermediate treatment. (**a**) FC, (**b**) WC.

### 3.2. High-Temperature Tensile Properties

Table 3 shows the results of the high-temperature tensile test of the high-Cr nickel-based superalloy with different solution temperatures. The tensile strength, yield strength, and elongation increase with the increase in solution temperature from 1160 °C to 1180 °C, then decrease when the temperature is higher than 1180 °C. In particular, the elongation shows a sharp decrease.

**Table 3.** The tensile test results at 900 °C of the superalloy with different solution temperatures.

| Heat Treatment | Solution Temperature/°C | Tensile Strength, $R_m$/MPa | Yield Strength, $R_{p0.2}$/MPa | Elongation, $A$/% |
|---|---|---|---|---|
| S1 | 1160 °C | 630.8 | 530.5 | 15.62 |
| S2 | 1180 °C | 683.6 | 592.5 | 16.95 |
| S3 | 1200 °C | 651.7 | 578.8 | 2.23 |
| S4 | 1220 °C | 538.1 | 394.3 | 0.55 |

The tensile test results at 900 °C of the superalloy with different cooling conditions after intermediate treatment are shown in Table 4. It can be seen that with the increase in cooling rates, the tensile strength slightly increases and then decreases, while the elongation decreases significantly.

**Table 4.** The tensile test results at 900 °C of the superalloy with different cooling conditions after intermediate treatment.

| Heat Treatment | Cooling Condition | Tensile Strength, $R_m$/MPa | Yield Strength, $R_{p0.2}$/MPa | Elongation, $A$/% |
|---|---|---|---|---|
| S5 | FC | 632.2 | 476.1 | 19.48 |
| S2 | AC | 683.6 | 592.5 | 16.95 |
| S6 | WC | 673.4 | 571.1 | 14.41 |

Figure 4 shows the longitudinal section microstructure of tensile fracture specimen heat-treated by S1, S2, S3, and S4 heat-treatment methods. It was shown that the cracks mainly initiated at the interface between the script-like MC-type carbide and matrix and propagated along the grain boundary and interdentritic region where the cast coarse $\gamma'$ phase had not completely dissolved into the matrix when the solution temperature was at 1160 °C (Figure 4a). As shown in Figure 4c,d, when the solution temperature reaches 1200 °C and above, the cracks usually form at the incipient regions and propagate along with the grain boundaries.

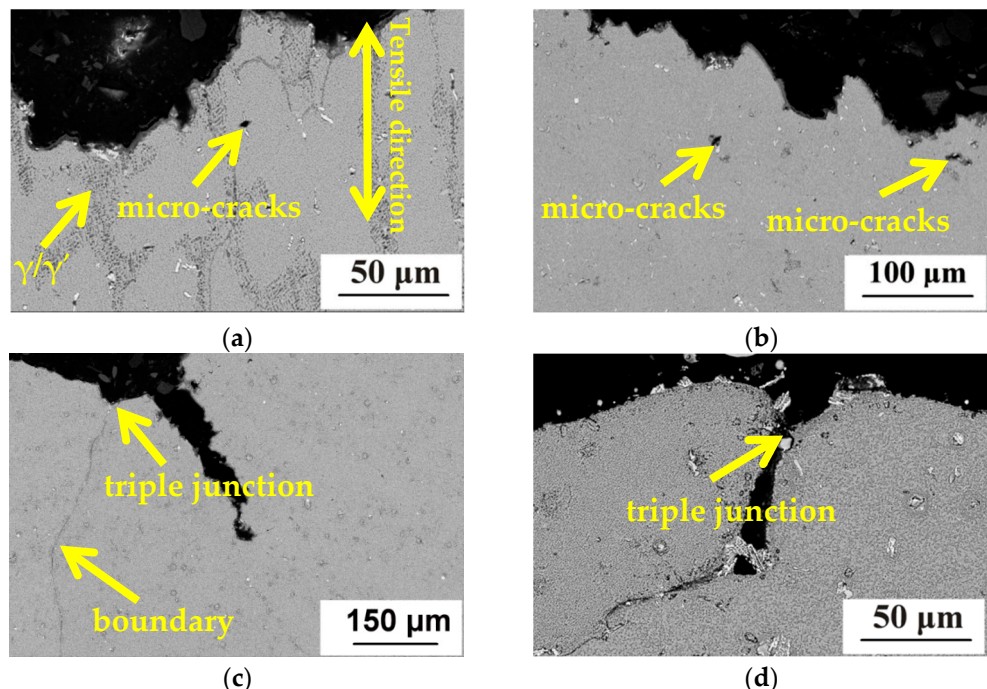

**Figure 4.** The longitudinal section microstructure of tensile fracture specimens with different solution temperatures. (**a**)1160 °C, (**b**) 1180 °C, (**c**) 1200 °C, (**d**) 1220 °C.

### 3.3. High-Temperature Rupture Properties

Table 5 shows the stress rupture results of the superalloy under the conditions of 900 °C/275 MPa. It can be seen that the elongation and reduction in area initially increase and then decrease with the increase in solution temperature. However, the fracture time gradually decreases with the increase in solution temperature. Compared with the high-temperature tensile test results, it can be seen that the variation trends with solution temperatures are similar.

**Table 5.** Stress rupture properties of the superalloy with different solution temperatures under conditions of 900 °C/275 MPa.

| Heat Treatment | Solution Temperature/°C | Rupture Time, t/h | Elongation, A/% | Reduction in Area, Z/% |
|---|---|---|---|---|
| S1 | 1160 °C | 142.06 | 8.55 | 3.97 |
| S2 | 1180 °C | 128.12 | 9.14 | 6.29 |
| S3 | 1200 °C | 36.29 | 2.83 | 5.52 |
| S4 | 1220 °C | 1.22 | 1.50 | 0.80 |

Table 6 shows the stress rupture properties of the superalloy with different cooling condition after intermediate treatment under condition of 900 °C/275 MPa. It can be seen that the fracture time, elongation, and reduction in area have no significant change with the increase of cooling rate.

**Table 6.** Stress rupture properties of the superalloy with different cooling conditions after intermediate treatment under conditions of 900 °C/275 MPa.

| Heat Treatment | Cooling Condition/°C | Rupture Time, t/h | Elongation, A/% | Reduction in Area, Z/% |
|---|---|---|---|---|
| S5 | FC | 123.61 | 9.03 | 5.14 |
| S2 | AC | 128.12 | 9.14 | 6.29 |
| S6 | WC | 123.62 | 7.03 | 7.84 |

## 4. Discussion

### 4.1. Effect of Solution Temperature on Alloy Properties

The microstructure of the high-Cr nickel-based superalloy consists of $\gamma$ phase matrix, $\gamma'$ precipitated phase, MC-type carbides and M6C-type carbides after the heat treatment, as shown in Figures 1 and 2. The carbides (MC, M6C) were distributed at grain boundaries and interdentritic regions. As shown in Table 3, the tensile strength, yield strength, and elongation increase with the solution temperature increasing from 1160 °C to 1180 °C. It can be analyzed from the following aspects. The high solution temperature makes the volume fraction of $\gamma/\gamma'$ eutectic gradually decrease and the mass volume fraction of $\gamma'$ particles increase. The higher mass fractions $\gamma'$ phase contribute to improving the strength of the superalloy during high-temperature stretching [3–7]. The high solution-treated temperature contributes to the element diffusion, and the decreased degree of segregation of the elements improves the uniformity of the alloy composition, beneficial to improving the mechanical properties [8].

However, when the solution temperature increases from 1180 °C to 1200 °C and 1220 °C, the tensile strength, yield strength, and elongation evidently decrease. The most important reason for this was that the morphology of carbides and $\gamma/\gamma'$ eutectic changed. Firstly, the high solution temperature contributes to the element diffusion and then leads to the grain boundaries carbides continuing to grow and connect, gradually becoming a small semi-continuous carbide chain and then growing into a thick continuous carbide chain [11–15]. For the diffusion of $\gamma'$-forming elements, the $\gamma'$ phase was precipitated around the grain boundaries of MC-type or M6C-type carbides and more coarsening, which further increased the grain boundary width [16,19,20]. Therefore, the micro-crack mainly formed at the grain boundaries carbides surrounding and propagated along grain boundaries during the high-temperature stretching, leading to the strength and elongation evidently decreasing.

At the same time, when the solution-treated temperatures were at 1200 and 1220 °C, the incipient melting appears in the interdendritic region, which causes the deterioration of mechanical properties [9,10].

As shown in Table 5, the 900 °C/275 MPa stress rupture tests curve, the elongations, and reduction in area initially increase and then decrease, and the fracture time gradually decreases. According to Figures 1 and 4, the microstructure shows that the grain boundaries carbides, the element diffusion, the mass volume fraction of $\gamma'$ phase, and the initial melting structure commonly lead to this curve change.

### 4.2. Effect of Cooling Rate on Alloy Properties

As shown in Table 4, the tensile strength initially increases and then decreases, and the elongation evidently decreases. It can be obtained from Figure 3 that with the increase in the cooling rate, the morphology of $\gamma'$ phase changes from cubic to spherical and precipitates the fine $\gamma'$ phase.

According to previous research, a higher mass volume fraction of $\gamma'$ phase will increase the properties of the alloy [11–13], and according to the deformation mechanism of dislocations, the dislocations will pass through the elliptical $\gamma'$ phase by a cutting mechanism and pass through the fine $\gamma'$ phase by an Orowan mechanism [14,15]. Hence, compared with the furnace cooling, the air cooling precipitating the fine $\gamma'$ phase is beneficial to improve the mechanical properties. So, when the tensile strength initially increases with increases the cooling rate.But adopts the water cooling, the high cooling rate will be beneficial to the $\gamma'$ phase nucleation and inhibit its growth, forming a large number of fine $\gamma'$ phases and leading to the decrease in mass volume fraction of $\gamma'$ phase and the smaller fine $\gamma'$ phase hindering dislocation movement, which was weakened [15,19–22]. So, compared with the air cooling, the water cooling in the tensile strength is decreased.

For the elongation of the alloy, we know that the smaller fine precipitated $\gamma'$ phase was beneficial to improve the hardness of the alloy and reduce the plasticity [11–13]. Hence, with the increase of cooling rate, the elongation evidently decreases.

According to previous research, the morphology of $\gamma'$ phase, fine $\gamma'$ phase, and carbides has no significant impact on the stress rupture test [19,20]. For the test process, the grain size plays an important role in the stress rupture properties; and because the different samples choose the same casting process, this leads to grain size similarity. So, the fracture time, elongation, and reduction in area have no significant change with the increase in cooling rate in the test at 900 °C/275 MPa.

## 5. Conclusions

In this paper, a type of high-Cr superalloy was designed and then different solution temperatures and intermediate treatment cooling conditions were used to tune the microstructure and properties. The microstructure and properties have been comprehensively characterized via SEM, EDS, high-temperature tensile tests and durable tests, and the main findings are summarized as follows:

(1) After the heat treatment, the microstructure of high-Cr superalloy consists of the carbides and $\gamma'$ precipitated phase and $\gamma/\gamma'$ eutectic. Among them, the carbides have an irregular distribution both inside the grains and on grain boundaries.

(2) With increasing solution-treated temperature, the tensile strength and the ductility initially increased and then decreased. The solution-treated temperature of the superalloy should not exceed 1180 °C; in order to achieve the optimal combination of strength and ductility, choose the heat treatment process: 1180 °C, 4 h (with air cooling) + 1050 °C, 14 h (with air cooling) + 750 °C, 12 h (with air cooling), where the tensile strength of the alloy is 683.6 MPa and the elongation is 16.95%.

(3) For the intermediate heat treatment used with different cooling methods, the morphology of the $\gamma'$ phase changed with the increased cooling rate. The morphology and distribution of $\gamma'$ phase have great influence on the tensile test results at 900 °C of the alloy but have no obvious influence on the test at 900 °C/275 MPa.

**Author Contributions:** Conceptualization, H.L.; methodology, H.L., M.Y., B.C. and F.Y.; validation, H.L., Z.M. and D.L.; formal analysis, F.Y. and L.Z.; writing—original draft preparation, H.L.; writing—review and editing, L.Z. and F.Y. All authors have read and agreed to the published version of the manuscript.

**Funding:** This research received no external funding.

**Data Availability Statement:** Not applicable.

**Conflicts of Interest:** The authors declare no conflict of interest.

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
