# Peer review of "Effects of Heat Treatment on the Microstructure and Properties of a Cast Nickel-Based High-Cr Superalloy"

_metals, doi:10.3390/met12122176_

Round 1
Reviewer 1 Report
1. Work hard on English grammar and style. Much work should be done to make the text understandable for the reader.
2. In the introduction, please explain the target alloy's composition and purpose. We should understand why You used these sources (articles) for the review and why it is necessary to focus on this point.
3. In the literature review, if You mention the alloy, please, indicate its chemical composition, at least as it is for [18, line 73]. The explained regularities are true for one alloy and false for the other.
4. High-chromium superalloys have been studied over the decades. I think You should browse the literature more carefully and analyze more articles before making the statement that they are not well-studied.
5. Please, in the "Materials..." section, specify whether this alloy is already used in the industry or You have fabricated a brand-new alloy
6. Line 94-95. What was the logic to select these regimes of heat treatment?
7. Line 116-120. In the article, high-resolution images and low-resolution images should be provided for all the samples. Otherwise, it is impossible to see how gamma-prime morphology changes.
8. Line120-121. Please, in Figure 1c, show the melting sites by arrows or circles (rectangles). Also, arrow the carbides and dendrides in Figure 1 ab. Is figure 1d a magnification from figure 1b?
9. Lines 128-130, Figure 2. There are no arrows in the Figure 2 ac
10. Lines 130-134. The EDS is not a reliable method in the identification of carbide type. For this, please, use XRD test
11. Figure 4. Please, add the picture showing the location of the image plane on the tensile sample and the direction of the tensile force application. The micro-cracks in Fig. 4 ab look like casting porosity. I think it is not a crack.
12. The headings 4 and 4.1 should be on the same page
13. Why didn't You do the solution treatment at 1900℃? Maybe at this temperature, the alloy performance is the best? I think the step of 200℃ is too big for this case. 100℃ is better.
14. Please, specify the cooling rate more accurately, in degrees/minute
15. Line 197-199. How did You measure γ/γ′ volume and mass fractions, and where in the manuscript are these data?
16. How does the cooling rate effects the formation of nanogammaprime particles in the gamma channel? Why do air and water-cooling at intermediate heat treatment result in nanogammaprime in the gamma channel, and why not after furnace cooling?
17. Line 255: chrysanthemum-like γ/γ′ eutectic: Please, add the picture with this type of microstructure to the article
18. Also, please follow the notions in the manuscript.
Best regards
Author Response
Response to reviewer #1:
We really appreciate you for your carefulness and conscientiousness. Your suggestions are really valuable and helpful for revising and improving our paper. According to your suggestions, we have made the following revisions on this manuscript:
- Work hard on English grammar and style. Much work should be done to make the text understandable for the reader.
Response: Thank you very much for your advice. I will strengthen the English grammar and style in the future.
- In the introduction, please explain the target alloy's composition and purpose. We should understand why You used these sources (articles) for the review and why it is necessary to focus on this point.
Response: Thank you very much for your advice. The target alloy's composition was shown in Table 1 and the purpose mainly apply to the gas turbines. When the gas turbines work under the complex environment, they often appear the high temperature phenomenon. It can damage the gas turbines work performance. Sowe carry out the high heat treatment on the target alloy and research the high temperature environment impact on the gas turbines.
Modified to: In this paper, a kind of high-Cr superalloy was designed, and contains many solid solution strengthening elements aiming at the gas turbines.
- In the literature review, if You mention the alloy, please, indicate its chemical composition, at least as it is for [18, line 73]. The explained regularities are true for one alloy and false for the other.
Response: Thank you very much for your advice. The alloy composition of 8% Cr、25%Mo and 62%Ni.
Modified to: Mao et al. [18] found the alloy composition of 8% Cr、25%Mo and 62%Ni.
- High-chromium superalloys have been studied over the decades. I think You should browse the literature more carefully and analyze more articles before making the statement that they are not well-studied.
Response: Thank you very much for your advice. According to many literatures and standards, most of the alloys during the high heat treatment temperature are below 1160℃. And the effect of high heat treatment temperature on high Cr superalloy precipitates has not been studied in detail.
- Please, in the "Materials..." section, specify whether this alloy is already used in the industry or You have fabricated a brand-new alloy
Response: Thank you very much for your advice. The target alloy was on the basis of K444 alloy, By adjusting the elements and controlling the harmful Elements, we form a new alloy. And the new alloy has excellent mechanical properties than K444 alloy.
Modified to: The nominal composition of the high-Cr superalloy alloy used in this study is listed in Table 1.The high-Cr superalloy was a improved alloy by K444.
- Line 94-95. What was the logic to select these regimes of heat treatment?
Response: Thank you very much for your advice. Because the alloy was on the basis of K444 alloy, the general heat treatment was invested. So we invest the high heat treatment on the alloy, and in order to further understand the alloy.
- Line 116-120. In the article, high-resolution images and low-resolution images should be provided for all the samples. Otherwise, it is impossible to see how gamma-prime morphology changes.
Response: Thank you very much for your advice. By comparing the difference of high-resolution images and low-resolution images, the low-resolution images are with little difference on morphology, and can’t be seen the gamma-prime morphology changes clearly.
- Line120-121. Please, in Figure 1c, show the melting sites by arrows or circles (rectangles). Also, arrow the carbides and dendrides in Figure 1 ab. Is figure 1d a magnification from figure 1b?
Response: Thank you very much for your advice. We have revised as per the advice. And the figure 1d was a magnification from figure 1b.
- Lines 128-130, Figure 2. There are no arrows in the Figure 2 ac
Response: Thank you very much for your advice. We have revised as per the advice.
- Lines 130-134. The EDS is not a reliable method in the identification of carbide type. For this, please, use XRD test
Response: Thank you very much for your advice.
According to previous studies, The title:Microstructure and mechanical properties of cast Ni-base superalloy K44;Title:Primary MC decomposition and its effects on the rupture behaviors in hot-corrosion resistant Ni-based superalloy K444;Title:Effects of solution cooling rate on microstructure and mechanical properties of nickel base superalloy K444. Through the EDS and previous paper, it can identify the carbide type.
- Figure 4. Please, add the picture showing the location of the image plane on the tensile sample and the direction of the tensile force application. The micro-cracks in Fig. 4 ab look like casting porosity. I think it is not a crack.
Response: Thank you very much for your advice. We have revised as per the advice. From the Fig.4 ab we can know that the micro-cracks are usually formed on the surface of carbiedes, and the casting porosity is often caused by casting defects. During the tensile, the casting porosity should be deformed by the direction of the tensile forec application. But in Fig.4 ab, the micro-cracks expansion is perpendicular to the tensile force.
- The headings 4 and 4.1 should be on the same page
Response: Thank you very much for your advice. We have revised as per the advice.
- Why didn't You do the solution treatment at 1900℃? Maybe at this temperature, the alloy performance is the best? I think the step of 200℃ is too big for this case. 100℃ is better.
Response:Thank you very much for your advice. I agree with the reviewer. We investied the target alloy by firstly choosing the solution treatment at 1160 ℃、1180 ℃、1200 ℃,1220 ℃.When finding the best heat treatment temperature, the after work was in small step by adjusting the solution temperature.
- Please, specify the cooling rate more accurately, in degrees/minute
Response: Thank you very much for your advice. I agree with the reviewer. If choose the vacuum heat treatment furnace, the cooling rate can be controlled more accurately. But the specimens were carried out in a muffle furnace, it can’t control the cooling rate accurately. And according to the industry standard, most of the casting alloy choose the air cooling. So we choose the furnace cooling (FC), air cooling (AC) and water cooling (WC) to invest the alloy.
- Line 197-199. How did You measure γ/γ′ volume and mass fractions, and where in the manuscript are these data?
Response: Thank you very much for your advice. According to many papers and from the Fig.1, we can know the γ/γ′ volume was decreased with the solutions temperature increasing At the same time, we can calculate by Image-pro plus software.
- How does the cooling rate effects the formation of nanogammaprime particles in the gamma channel? Why do air and water-cooling at intermediate heat treatment result in nanogammaprime in the gamma channel, and why not after furnace cooling?
Response: Thank you very much for your advice. Because the furnace cooling (FC), air cooling (AC) and water cooling (WC) cooling rate was different. When choose the furnace cooling, due to low cooling rate, the nano gamma phase can have enough time to coarsen and grow up.
- Line 255: chrysanthemum-like γ/γ′ eutectic: Please, add the picture with this type of microstructure to the article
Response: Thank you very much for your advice. We have revised as per the advice.The chrysanthemum-like γ/γ′ eutectic on the Fig.1.
- Also, please follow the notions in the manuscript.
Response: Thank you very much for your advice.

Reviewer 2 Report
The paper deals with the current issue of heat treatment of nickel-based superalloys.
The reviewer appreciates the effort put into the research and the documentation of the results. At the same time, the noted comments and some critiques will help to improve the quality and scientific value of the manuscript.
The written text should once again be reviewed and checked for style. The reviewer noted repeated and redundant words, lack of a capital letter at the beginning of a sentence, incorrectly used hyphens, etc.
The sentence 'As we all know, the microstructure of nickel-based superalloy consists of the γ′ precipitates, carbides, and γ/γ′ eutectic structure' (line 43) is not entirely true, as borides, nitrides, TCP phases and other can appear in Ni-based superalloys. It would need to be added whether this sentence applies to all Ni superalloys or polycrystalline (carbides are generally not present in monocrystalline).
There is slight confusion in the literature review. The introduction provides only generalized background of the topic. There is no information on the effect of heat treatment time on the results in terms of the desired superalloy microstructure.
No explanation for why such and not other heat treatment temperatures were adopted in the tests.
What is the practical application of the results obtained with 'The intermediate heat-treatments were followed by furnace cooling (FC), air cooling (AC) and water cooling (WC), respectively. All heat treatment experiments were carried out in a muffle furnace and under the air atmosphere' (line 96) when the heat treatment of Ni superalloys is customarily carried out under a protective gas atmosphere and at a high cooling rate?
Why were the conditions of the tensile test chosen as stated? What determined the choice of such parameters?
The text mentions that Figure 1 shows the results of S1, S2 and S3 (line 113). The caption of Figure 1 indicates the effects of applying various solution treatment temperatures, but image 'd' is signed as the result of applying S2, i.e. not only after solid solution treatment but also after intermediate treatment. Was this the intention of the authors? The present caption and text raise doubts for the reader.
Arrows A and B described in the text (line 130-132) do not appear in the illustrations.
Merely finding the presence of an element is not an infallible guideline for claiming the type of carbide (lines 131-132, 193). No other test confirming the type of carbide described.
The text in lines 140-141 is not consistent with the caption of Figure 3.
In lines 164-166 there is written 'It is shown that the cracks mainly initiate at the interface between the script-like MC-type carbide and matrix, and propagate along the interdendritic region where the cast coarse γ′ particles have not completely dissolved into matrix when the solution temperature is at 1160 ℃ (Fig. 4a)'. The statement about crack propagation along interdendritic areas is not entirely true when looking at Figure 4a.
The text (line 167) mentions figure 5, but there is no such figure after all.
The term 'chrysanthemum-like γ/γ′ eutectic' (line 255) can be misleading and not entirely true.
The claim of optimum plasticity (line 259) should be related to the test results obtained. What does optimum ductility mean in the case under consideration?
Author Response
Response to reviewer #2:
We really appreciate you for your carefulness and conscientiousness. Your suggestions are really valuable and helpful for revising and improving our paper. According to your suggestions, we have made the following revisions on this manuscript:
1.The paper deals with the current issue of heat treatment of nickel-based superalloys.
Response: Thank you very much for your advice.
- The reviewer appreciates the effort put into the research and the documentation of the results. At the same time, the noted comments and some critiques will help to improve the quality and scientific value of the manuscript.
Response: Thank you very much for your advice.
3.The written text should once again be reviewed and checked for style. The reviewer noted repeated and redundant words, lack of a capital letter at the beginning of a sentence, incorrectly used hyphens, etc.
Response: Thank you very much for your advice.
4.The sentence 'As we all know, the microstructure of nickel-based superalloy consists of the γ′ precipitates, carbides, and γ/γ′ eutectic structure' (line 43) is not entirely true, as borides, nitrides, TCP phases and other can appear in Ni-based superalloys. It would need to be added whether this sentence applies to all Ni superalloys or polycrystalline (carbides are generally not present in monocrystalline).
Response:Thank you very much for your advice. We have revised as per the advice.
As we all know, the microstructure of nickel-based superalloy consists of the γ′ precipitates, carbides, and γ/γ′ eutectic structure.
The sentence was modified to:As we all know, the microstructure of casting nickel-based superalloy consists of the γ′ precipitates, carbides, borides, nitrides, TCP phases, γ/γ′ eutectic structure and other precipitated phase.
5.There is slight confusion in the literature review. The introduction provides only generalized background of the topic. There is no information on the effect of heat treatment time on the results in terms of the desired superalloy microstructure.
Response: Thank you very much for your advice. The paper was mainly to focus on the solutions temperature and cooling rate influence on the alloy. So in the literature review the description of heat treatment time information was little.
6.No explanation for why such and not other heat treatment temperatures were adopted in the tests.
Response: Thank you very much for your advice.
Due to few people have studied the effect of high temperature solution treatment on the mechanical properties of alloy. Hence, we choose this work. At the same time, the gas turbines in use often appears the high temperature heat treatment phenomenon, therefore these works can be provided for research reference.
- What is the practical application of the results obtained with 'The intermediate heat-treatments were followed by furnace cooling (FC), air cooling (AC) and water cooling (WC), respectively. All heat treatment experiments were carried out in a muffle furnace and under the air atmosphere' (line 96) when the heat treatment of Ni superalloys is customarily carried out under a protective gas atmosphere and at a high cooling rate?
Response: Thank you very much for your advice. According to the actual condition, most of cast superalloys heat treatments adopt the muffle furnace, except for few alloy adopting the vacuum. In this paper, the alloy heat treatments adopt the professional standard by Q/703J 178-2016. Hence, according to the standard, we choose the muffle furnace. There is no need the protective gas atmosphere.
8.Why were the conditions of the tensile test chosen as stated? What determined the choice of such parameters?
Response: Thank you very much for your advice. In order to understand the alloy, we firstly choose the different solutions temperature, and select the best heat treatment temperature. After that, we carry out the 900℃ tensile and durable test. The 900℃ tensile and durable test was referred to the industry standard(Q/703J 178-2016).
9.The text mentions that Figure 1 shows the results of S1, S2 and S3 (line 113). The caption of Figure 1 indicates the effects of applying various solution treatment temperatures, but image 'd' is signed as the result of applying S2, i.e. not only after solid solution treatment but also after intermediate treatment. Was this the intention of the authors? The present caption and text raise doubts for the reader.
Response: Thank you very much for your advice. The figure 1 shows the results of S1, S2 and S3. Because different solution temperatures have no obvious impact on γ′ particles, we just apply on the picture d. The other reason to choose the picture d is to compare with the Figure 3 so that let readers can understand that different cooling rates have a big impact on the γ′ particles.
10.Arrows A and B described in the text (line 130-132) do not appear in the illustrations.
Response: Thank you very much for your advice. We have revised as per the advice.
11.Merely finding the presence of an element is not an infallible guideline for claiming the type of carbide (lines 131-132, 193). No other test confirming the type of carbide described.
Response: Thank you very much for your advice.
The target alloy was on the basis of K444 alloy, through the elements fine adjust and control the harmful Elements, its actions can contribute to improve the properties of mechanical, and have excellent mechanical properties than K444 alloy. But the precipitated phase does not change.
According to previous studies, The title:Microstructure and mechanical properties of cast Ni-base superalloy K44;Title:Primary MC decomposition and its effects on the rupture behaviors in hot-corrosion resistant Ni-based superalloy K444;Title:Effects of solution cooling rate on microstructure and mechanical properties of nickel base superalloy K444.
So we can know that the two type carbides were MC and M6C.
12.The text in lines 140-141 is not consistent with the caption of Figure 3.
Response: Thank you very much for your advice. We have revised as per the advice.
13.In lines 164-166 there is written 'It is shown that the cracks mainly initiate at the interface between the script-like MC-type carbide and matrix, and propagate along the interdendritic region where the cast coarse γ′ particles have not completely dissolved into matrix when the solution temperature is at 1160 ℃ (Fig. 4a)'. The statement about crack propagation along interdendritic areas is not entirely true when looking at Figure 4a.
Response: Thank you very much for your advice. I Agree with the reviewer. The script-like carbides are distributed in grain boundaries and interdendritic region. So during the high temperture tensile, the crack is propagated along interdendritic and gain boundary.
Modified to: It was shown that the cracks mainly initiate at the interface between the script-like MC-type carbide and matrix, and propagate along the grain boundary and interdentritic region where the cast coarse γ′ phase have not completely dissolved into matrix when the solution temperature was at 1160 ℃ (Fig.4a)
14.The text (line 167) mentions figure 5, but there is no such figure after all.
Response:Thank you very much for your advice. We have revised as per the advice.
15.The term 'chrysanthemum-like γ/γ′ eutectic' (line 255) can be misleading and not entirely true.
Response:Thank you very much for your advice. We have revised as per the advice.
Line262: chrysanthemum-like γ/γ′ eutectic.Modified to:γ/γ′ eutectic.
16.The claim of optimum plasticity (line 259) should be related to the test results obtained. What does optimum ductility mean in the case under consideration?
Response: Thank you very much for your advice. The plasticity and ductility were reflect properties of a material. When we did the tensile test, only the elongation was recorded, and the reduction in area was for reference. So in this paper, we haven’t discussed it any further.

Reviewer 3 Report
The present manuscript is dealing with the microstructure and mechanical properties of a cast Nickel-based superalloy after different heat treatments. The detailed effects of solution treatments and cooling rate on the evolution of microstructure, tensile and stress rupture properties have been investigated and discussed. Some valuable results have been obtained which contributes to the research of Nickel-based superalloys with high Cr addition. However, the paper should go through careful and thorough corrections before the acceptance.
1. In the Abstract, we consider that incipient melting occurs in the interdendritic region of alloy rather than “incipient melting structure appears”. It is recommended to change and modify the corresponding statement throughout the manuscript.
2. In the Materials and Methods part, paragraph 2, the writing of the upper and lower targets should be standardized, it is recommended to carefully check the full text.
3. In section 3.1, The authors stated that the intermediate treatment cooling rate had an important effect on γ′ phase. Could the authors explain the detail of this phenomenon?
4. In section 3.1, some papers have reported that MC carbides containing Ti and W decomposed into M6C carbides containing W and Mo during the solution treatment (J Mater Sci (2006) 41:6476). While the selective area diffraction pattern of the carbides would be beneficial to confirm the transition.
5. In section 3.1, figure 1 and figure 2, some small particles could be observed along the grain boundaries, which might be also considered as M23C6 particles or borides according to the morphology characteristics. The authors are suggested to reconsider this phenomenon and make the research results more convincing.
6. In section 3.2, the tensile properties of Table 3, and Table 4 should be organized into graphs, and the comparison will be more clear and more comparable.
7. In section 3.2, the elongation of S3 sample decreased sharply compared to the S2 sample, could the author explain this phenomenon at length?
8. In section 4.1, paragraph 4, when the solution treatment temperature exceeds the incipient melting temperature, the mechanical properties of superalloys decrease incredibly. It is generally known that the solution treatment temperature should be lower than the incipient melting temperature, why did these specimens go through the subsequent tensile and stress rupture testing?
9. In section 4.2, paragraph 2, actually, the plasticity of superalloys could not increase with the decrease of γ′ phase size without limitation. The authors should be more careful to give conclusions.

Author Response
Response to reviewer #3:
We really appreciate you for your carefulness and conscientiousness. Your suggestions are really valuable and helpful for revising and improving our paper. According to your suggestions, we have made the following revisions on this manuscript:
1.In the Abstract, we consider that incipient melting occurs in the interdendritic region of alloy rather than “incipient melting structure appears”. It is recommended to change and modify the corresponding statement throughout the manuscript.
Response: Thank you very much for your advice. We have revised as per the advice.
Line23:the incipient melting structure appears in the matrix.Modified to:the incipient melting appears in the interdendritic region.
Line127:the matrix appears incipient melting structure.Modified to:the interdendritic region appears incipient melting.
Line224:the incipient melting appears in the matrix which causes the deterioration of mechanical properties.
Modified to:the incipient melting appears in the interdendritic region which causes the deterioration of mechanical properties.
2.In the Materials and Methods part, paragraph 2, the writing of the upper and lower targets should be standardized, it is recommended to carefully check the full text.
Response: Thank you very much for your adviceWe have revised as per the advice.
- In section 3.1, The authors stated that the intermediate treatment cooling rate had an important effect on γ′ phase. Could the authors explain the detail of this phenomenon?
Response: Thank you very much for your advice.When we choose different intermediate treatment cooling rates, the target alloy on the heat treatment time will be greatly different.The furnace cooling make γ′ phase have the enough time to grow up, and form the cubic morphology.
4.In section 3.1, some papers have reported that MC carbides containing Ti and W decomposed into M6C carbides containing W and Mo during the solution treatment (J Mater Sci (2006) 41:6476). While the selective area diffraction pattern of the carbides would be beneficial to confirm the transition.
Response: Thank you very much for your advice. According to the figure 2 of BSE picture, we select the area on the carbides to do analysis of element composition. So we can confirm that the carbides containing W is M6C.
- In section 3.1, figure 1 and figure 2, some small particles could be observed along the grain boundaries, which might be also considered as M23C6 particles or borides according to the morphology characteristics. The authors are suggested to reconsider this phenomenon and make the research results more convincing.
Response: Thank you very much for your advice. According to the previous SEM observation and energy spectrum analysis, we can’t found the borides along the grain boundaries. So we analyse the microstructure of the alloy is mainly composed of γ matrix, ordered γ′phase, and carbides.
- In section 3.2, the tensile properties of Table 3, and Table 4 should be organized into graphs, and the comparison will be more clear and more comparable.
Response: Thank you very much for your advice.When the tensile properties were organized into graphs, the details of tensile properties could be ignored. Sot we choose the form, such as Table 3, and Table 4.
7.In section 3.2, the elongation of S3 sample decreased sharply compared to the S2 sample, could the author explain this phenomenon at length?
Response: Thank you very much for your advice. When the target alloy adopts the heat treatment of S3,the high solutions temperature will lead to incipient melting in the interdendritic region, which causes the deterioration of mechanical properties.
- In section 4.1, paragraph 4, when the solution treatment temperature exceeds the incipient melting temperature, the mechanical properties of superalloys decrease incredibly. It is generally known that the solution treatment temperature should be lower than the incipient melting temperature, why did these specimens go through the subsequent tensile and stress rupture testing?
Response: Thank you very much for your advice. I agree with the reviewer. Because the target alloy was an improved alloy, which aims to clearly understand the mechanical properties, so we design the high solutions treatments.
9.In section 4.2, paragraph 2, actually, the plasticity of superalloys could not increase with the decrease of γ′ phase size without limitation. The authors should be more careful to give conclusions.
Response: Thank you very much for your advice. I agree with the reviewer.

Round 2
Reviewer 1 Report
Dear Authors!
Thank You for the work done.
Please, pay attention to Figure 2. The arrow mowed away from picture C. The word "Point" is not necessary
Author Response
1.Please, pay attention to Figure 2. The arrow mowed away from picture C. The word "Point" is not necessary
Response: Thank you very much for your advice.We have revised as per the advice.

Reviewer 2 Report
The reviewer thanks the authors for their cooperation and for taking suggestions into account.
Author Response
Thank the reviewer.

Reviewer 3 Report
The authors have made some corrections to improve the quality of the manuscript. However, there are still some issues ignored by the authors.
1. We have not found the cover letter in the current form, please submit the cover letter again.
2. We mentioned the method of TEM observation and SADP analysis to determine the carbides. However, the authors gave no response!
3. It is better to draw the strain-stress curves and tensile property curves, rather than list the tensile properties in the forms. The strain-stress curves are most commonly used to illustrate the tensile behaviors of the materials.
4. The English language needs to be improved further, please ask for help from the professional institute or native speakers. The authors should pay attention to this problem and improve the quality of the manuscript!
Author Response
Response to reviewer #3:
We really appreciate you for your carefulness and conscientiousness. Your suggestions are really valuable and helpful for revising and improving our paper. According to your suggestions, we have made the following revisions on this manuscript:
1.We have not found the cover letter in the current form, please submit the cover letter again.
Response: Thank you very much for your advice. Sorry to the reviewer for any inconvenience caused. The relevant problems have been corrected.
2.We mentioned the method of TEM observation and SADP analysis to determine the carbides. However, the authors gave no response!
Response: Thank you very much for your advice. Yes,it is better to determine the carbides using the TEM observation method. However, we think it can be identified according to their morphology, energy spectrum analysis and the BSE images in Figure 2 because those carbides in Ni-base superalloys have been fully discussed in literatures.
- It is better to draw the strain-stress curves and tensile property curves, rather than list the tensile properties in the forms. The strain-stress curves are most commonly used to illustrate the tensile behaviors of the materials.
Response: Thank you very much for your advice. We agree that the strain-stress curves are better to illustrate the tensile behaviors of the materials.
Unfortunately, the tensile properties results were obtained from the third party testing institutions, the original data is not provided.
- The English language needs to be improved further, please ask for help from the professional institute or native speakers. The authors should pay attention to this problem and improve the quality of the manuscript!
Response: Thank you very much for your advice. We have revised as per the advice.
Line12:a new cast nickel-based high-Cr superalloy were investigated in this paper.
Modified to:a new cast nickel-based high-Cr superalloy was investigated in this paper.
Line25:a large number of fine γ' phase precipitation in the γ channel was found with increaseing cooling rate after intermediate heat-treatment.
Modified to:a large number of fine γ' phase precipitates in the γ channel are found with increasing cooling rate after intermediate heat-treatment.
Such as :
Line34,68,72,74,75,78,79,87,88,91,100,123,137,144,150,153,172,173,174,222,232,247,251,252,274
was changed.

Round 3
Reviewer 3 Report
The current form can be accepted for publication.